# Difficulties in Establishing the Adverse Effects of β-Casomorphin-7 Released from β-Casein Variants—A Review

**DOI:** 10.3390/foods12173151

**Published:** 2023-08-22

**Authors:** Marta Liliane de Vasconcelos, Luisa Maria F. S. Oliveira, Jeremy Paul Hill, Ana Maria Centola Vidal

**Affiliations:** 1Department of Veterinary Medicine, Faculty of Animal Science and Food Engineering, University of São Paulo, Pirassununga 13635-900, SP, Brazil; martavasconcelos@usp.br (M.L.d.V.); luisamariafso@usp.br (L.M.F.S.O.); 2Department Sustainable Nutrition Initiative, Riddet Institute, Palmerston North, New Zealand, and Fonterra Research & Development Centre, Palmerston North 4472, New Zealand; jeremy.hill@fonterra.com

**Keywords:** β-Casomorphin, casein, genotype, lactose intolerance, milk, opioid

## Abstract

β-Casomorphin-7 (BCM-7) is a peptide released through the proteolysis of β-casein (β-CN), which is considered a bioactive peptide displaying evidence of promoting the binding and activation of the μ-opioid receptor located in various body parts, such as the gastrointestinal tract, the immune system and potentially the central nervous system. The possible effects of BCM-7 on health are a theme rising in popularity due to evidence found in several studies on the modulation of gastrointestinal proinflammatory responses that can trigger digestive symptoms, such as abdominal discomfort. With the advancement of studies, the hypothesis that there is a correlation of the possible effects of BCM-7 with the microbiota–gut–brain axis has been established. However, some studies have suggested the possibility that these adverse effects are restricted to a portion of the population, and the topic is controversial due to the small number of in vivo studies, which makes it difficult to obtain more conclusive results. In addition, a threshold of exposure to BCM-7 has not yet been established to clarify the potential of this peptide to trigger physiological responses at gastrointestinal and systemic levels. The proportion of the population that can be considered more susceptible to the effects of BCM-7 are evidenced in the literature review. The challenges of establishing the adverse effects of BCM-7 are discussed, including the importance of quantifying the BCM-7 release in the different β-CN genotypes. In summary, the reviewed literature provides plausible indications of the hypothesis of a relationship between β-CN A1/BCM-7 and adverse health effects; however, there is need for further, especially in vivo studies, to better understand and confirm the physiological effects of this peptide.

## 1. Introduction

Milk and dairy products are great nutritional sources because they provide essential macronutrients and micronutrients. A study that evaluated the global supply of nutrients to the human population found that milk contributes to 28 of the 29 nutrients included in the human nutritional needs [1]. Milk is the largest single source of protein for the human population, releasing a variety of peptides and essential amino acids of high biological value [1,2]. The main constituent of milk protein is casein, which accounts for around 80% of its content, and is a well-known source of bioactive compounds of various organic actions [3,4]. β-casein (β-CN) represents over 30% of total casein and is gaining importance in nutrition science due to its potential to release bioactive peptides, some of which have opioid characteristics [5].

Much has been discussed about the hypothesis that milk containing A1 β-CN or equivalent genetic variants has the potential to release bioactive peptide β-Casomorphin-7 (BCM-7) in a sufficient amount to trigger adverse health effects through the opioid pathway [6,7,8,9]. On the other hand, it is believed that milk containing only A2 β-CN is not associated with adverse health effects due to the low release of BCM-7.

BCM-7 is considered a peptide with potential to affect the gut–brain axis through immunoinflammatory responses related to the gastrointestinal system [10]. Although there is extensive literature on the action of BCM-7, many of these studies were performed in vitro or with animal models, as human analysis is more difficult and expensive [11,12], which is why the hypothesis in relation to β-CN A1/BCM-7 remains controversial and inconclusive.

Critical questions still need to be answered, such as the following: Which individuals in the population and in which health conditions are most susceptible to the effects of BCM-7? At which BCM-7 levels will these effects become apparent and how do they vary between individuals and population subgroups? Similarly, the elimination rate of BCM-7 may be an important factor in determining possible adverse biological effects, as no specific amount of BCM-7 has been designated as the minimum load of relevance to trigger or influence physiological responses.

In this context, the aim of this review is to increase the understanding related to these issues through a critical analysis of the literature, including in vitro and in vivo clinical trials, both in animals and humans.

## 2. β-Casein Polymorphism

The cow milk proteins, as well as milk proteins from other mammals (goat, sheep, buffalo and camel), can be divided into two groups: caseins (CNs), which constitute approximately 80% of the total protein content, and whey proteins, representing 20% [4,13]. Due to their high technological and nutritional representativeness and functionality, caseins have been extensively studied, including structural studies, initially as models of casein micelles, first proposed by Waugh in 1958, up to the most recent molecular analysis studies, such as the micellar model of CNs in nanocluster defined by Kruif and Holt (2003) [14].

Caseins are classified as phosphoproteins and characterized into four isoforms: alpha_S1_-casein (α_S1_-CN), alpha_S2_-casein (α_S2_-CN), beta-casein (β-CN) and kappa-casein (k-CN), organized in a micellar format according to their electrophobic interactions. CN phosphoproteins are arranged together according to their hydrophobicity, electrostatic interactions and interact with minerals collectively; they are referred to as casein-calcium-phosphate (CCP) [4].

In the association of the four different supramolecular structures of α_S1_-CN, α_S2_-CN, β-CN and k-CN, the elements are arranged in different amounts, in proportions of approximately 26%, 8%, 30% and 15%, respectively. In addition, the structural characteristics and the amount of amino acids of the CNs can be influenced by the genetic variation of casein proteins [15,16]. Casein genes have been extensively studied in bovine species; the casein region is located on chromosome 6 (BTA6q31). The genes have high mutation rates, especially in the β-CN coding [17,18,19].

Investigations into CN protein polymorphisms that allowed the identification of genetic variations began more than 50 years ago and confirmed the high rate of evolution of the four (4) genes, but at the same time showed that the overall organization of genes was conserved [20]. The high mutation rate mainly of β-CN is justified by the rapid evolution of the CSN2 gene due to the minimum structural requirements for function, so the alignment of sequences allows variation in a specific position of the DNA (SNP). This modifies the AA sequence of the encoded protein, resulting in nucleotide substitutions that lead to the exchange of protein amino acids called missense mutation, the variation of which is the most frequent form of polymorphism [20,21].

Since the identification of β-CN by Aschaffenburg (1968) [22], variants have also been identified. Knowledge about β-CN variants was expanded between 1965 and 1970, when gel electrophoresis methods were first developed, so the identified A-type variants were separated into A1, A2 and A3 [23]. The β-CN protein is a 209-amino-acid-long peptide chain, with thirteen (13) identified genetic variants (A1, A2, A3, A4, B, C, D, E, F, G, H1, H2 and I), resulting from changes in the 209 amino acids in the sequence. These genetic variations have characteristics of codominance, which means that alleles may be present in either homozygous, i.e., A1A1 or A2A2, etc., or heterozygous, i.e., A1A2, A1B, etc., forms, where both alleles are expressed for each of the 13 variants [24].

The A2 type variant appears to be the original form of β-CN, as it is the most common, with high frequency among cattle breeds [23], with even higher prevalence in buffalo breeds, since in buffalos, the CSN2 gene is similar to the bovine sequence [25,26,27]. In other dairy species, the similarity of the CSN2 gene and the coding of the A2 allele from β-CN have not yet been clearly established. Furthermore, the second highest prevalence comes from the A1 variant in cattle, classified as a product of the mutation by natural selection. The high prevalence of both A1 and A2 variants found in the dairy herd is responsible for most commercial bovine milk generally containing a mixture of A1 and A2 β-CN. A1A2 and A2A2 genotypes are the most frequent, while other genotypes are considered rare or of very low prevalence, such as the A1A1 genotype [23].

The genetic difference between A1 and A2 of β-CN alleles is the result of the polymorphism of a single codon in the β-CN gene, where cytosine is replaced by adenine, resulting in change in the protein sequence from proline at position 67 (Pro^67^) to histidine (His^67^) [28,29].

When milk presents β-CN expressing the A1 variant, homozygous or heterozygous (A1A1/A1A2), this exchange of a single amino acid His^67^ facilitates cleavage by proteases. Cleavage occurs in the small intestine by the enzymatic action of pepsin and leucine aminopeptidase, releasing the tyrosine residue in the amino terminal by cleaving the Val^59^-Tyr^60^ peptide, while pancreatin releases the terminal carboxyl, cleaving the Ile^66^-His^67^ site of A1 β-CN [30]. The process releases several amino acids, among them β-Casomorphin-7 (BCM-7), a bioactive peptide with a seven amino acid long chain (Tyr^60^-Pro^61^-Phe^62^-Pro^63^-Gly^64^-Pro^65^-Ile^66^) with opioid characteristics (Figure 1) [31].

In the A2 variant, the peptide bond between isoleucine and proline has more enzymatic resistance, which makes it difficult for proteases to proceed the cleavage between positions 66 and 67; on the other hand, chain cleavage occurs in the nine amino acid long peptide known as β-Casomorphin-9 (BCM-9) (Tyr^60^-Pro^61^-Phe^62^-Pro^63^-Gly^64^-Pro^65^-Ile^66^-Pro^67^-Asn^68^) (Figure 1), which is considered a potentially bioactive peptide with antihypertensive and antioxidant properties [31,32].

## 3. Digestion of β-CN and Generation of BCM-7

There are a considerable number of bioactive peptides released after the digestion of proteins from dairy products with antimicrobial, antioxidant, antihypertensive and immunomodulatory activity [33,34]. In general, bioactive peptides released from bovine milk proteins are beneficial to human health. One of the main challenges of this statement is to prove their efficacy and potential for in vivo bioavailability since their activity depends on gastrointestinal digestion and ability to reach target organs [34]. Several studies have focused on strengthening and confirming this hypothesis, and as advanced studies have been carried out, it is possible to understand that the structural integrity of some peptides is not compromised at gut level, as well as their transport through the epithelium, justified by structural differences, characteristic of hydrophobicity, size, molecular weight and peptide charge [33].

All β-Casomorphins released by the proteolysis of β-CN contain 4 to11 amino acids, and the first 3 amino acids of the chain remain identical: tyrosine-proline-phenylalanine (Tyr-Pro-Phe), starting at position 60 from the sequence of the 209 β-CN amino acids (Table 1). This means that the main difference between these peptides classified as β-Casomorphin is the amount of amino acids, determined by the subsequent cleavage to position 3, which consequently results in the release of the eight types of β-Casomorphin (Table 1) [35].

In humans, casein digestion starts in the stomach, where enzymes hydrolyze peptides of various lengths, continues in the duodenum, and, finally, proceeds to jejunal digestion, where hydrolysis and the absorption of smaller peptides occurs [10]. For the absorption of proteins in the intestine, it is necessary that these proteins break into amino acids or small peptides, for example, dipeptides and tripeptides [6]. Physiologically, dipeptidyl peptidase IV (DPP-IV) is the main enzyme responsible for the hydrolysis of dipeptides into free amino acids [36], a cell surface enzyme present in various types of cells, such as endothelial cells and brush border in the intestinal mucosa, in various intestinal bacteria strains, which are present in the soluble form in the bloodstream [2,37]. In fact, DPP-IV is considered the main enzyme responsible for the cleavage of BCM-7, as it selectively removes the dipeptide that contains proline (Pro^61^) in N-terminal peptides [16] (Figure 2). Therefore, the study of this interaction is essential.

The identification of BCM-7 in the human body and its potential impacts have attracted the attention of the scientific community. However, β-CN releases BCM-7, which occurs mainly after the hydrolysis of β-CN A1. BCM-7 is classified as an exorphin because it is an exogenous opioid peptide, with the same classification as that of morphine [38].

Since BCM-7 is released into the intestinal lumen in about 30 min to 6 h after ingestion of β-CN and because it is an exorphin, numerous studies show concern about its physiological activity as an opioid agonist, potential to activate type μ opioid receptor [39]. These receptors are located in the Central Nervous System (CNS) and the peripheral system, throughout the gastrointestinal tract, bladder, and in the cells of the immune and endocrine system [40].

This relationship of possible effects of BCM-7 on human physiological function is motivated by its structural similarity with other opioids, allowing its interaction with μ-opioid receptors, such as enkephalins and endorphins, which are endogenous opioid neurotransmitters that interact with μ-opioid receptors [41]. This affinity in the μ-opioid receptor active site is due to the presence of Tyr at its end in the N-terminal as a morphine- like endogenous peptide, considering that the amino acid sequence of these peptides has the same structural characteristic. The opioid system has the following regulatory and control functions: inhibition of pain stimuli, endocrine and autonomic nervous system functions, emotions and cognitive ability, learning and modulation of memory and gastrointestinal functions [8,10].

Therefore, when BCM-7 binds to the active site of μ-opioid receptors, it is assumed that it is also capable of causing structural modifications that activate signal transduction, generating biological responses [10]. The main potential agonist effect of opioid BCM-7 is the modulation of motility that delays gastrointestinal transit time and increases mucus production in humans [40].

BCM-7 can pass through the intestinal barrier and enter the bloodstream [16]. Some aspects remain unclear; for example, whether BCM-7 has the potential to modulate responses by binding to μ-opioid receptors outside the gastrointestinal tract, as the cause and effect associated with dose limits and duration/exposure of BCM-7 consumption in the body of humans, which is a factor that can modulate physiological functions.

The potential physiological effect of BCM-7 in animals and humans has been widely studied. However, most studies focus on inflammatory gastrointestinal responses, as well as the modulation of responses linked to opioid receptors in the nervous and immune systems [42]. It should be highlighted that milk commonly marketed worldwide contains both β-CN A1 and A2 variants, and the health benefits of milk are valid for the entire world population.

## 4. BCM-7 Threshold on Physiological Effects

The detection of BCM-7 can be quantified in gastrointestinal fluids, blood and urine by techniques combined with high-performance liquid chromatography (HPLC) and the ELISA kit [11,43]. Although some studies have shown the release of BCM-7 after enzymatic hydrolysis of A2 β-CN, the amount found of this peptide is lower compared to that found after enzymatic hydrolysis of A1 β-CN; therefore, the low amount of BCM-7 that can be exclusively released from A2 milk is considered irrelevant or low for causing effects on human health [44]. However, the threshold of exposure to BCM-7 with the potential to cause physiological effects is not yet clear in the literature.

Ramakrishnan et al. [45] elucidated this hypothesis by studying the effect of different proportions of β-CN variants on milk, monitoring consumption by individuals with health conditions associated with comorbidities. Symptoms associated with gastrointestinal discomfort were correlated with the consumption of A2A2 milk, milk from Jersey A1A2 cows (25% of A1 variant and 75% of A2 variant) and conventional A1A2 milk (75% A1 and 25% A2). As a result, it was observed that both the consumption of Jersey milk and the consumption of conventional milk did not reduce abdominal pain, bloating, flatulence and diarrhea symptoms, which are usually related to milk consumption, while the consumption of A2A2 milk was significantly (*p* = 0.004) associated with the reduction in abdominal pain. This result suggests that even a small amount of A1 β-CN can trigger gastrointestinal discomforts. It is evident that to confirm this hypothesis, further studies are required, including larger populations and the association with the release rate of BCM-7, as well as exposure time.

Such studies are difficult to perform and have a high cost of technologies involved, require greater complexity and robustness, and are time consuming, in addition to the use of exploratory methods that lead to increased stress in these individuals [46]. Some studies have correlated the effects of bioactive opioid peptides on humans, clinical blood examination or secondary symptomatology in abdominal regions and stool consistency [7,47]. Other studies have used a simplified digestion in in vitro, ex vivo models and animal in vivo model as alternatives, simulating the digestion process of humans [46].

In the study by Asledottir et al. [3], the difference in the release rate of BCM-7 between genotypes in ex vivo digestion exceeded the proportion found in previous studies after the digestion of A1A1 and A2A2 milk. The amount of BCM-7 released after 1 h of A1A1 milk digestion was 1.85 mg/g β-CN, which corresponds to 3.71 mg/200 mL of milk. After the digestion of A2A2 milk, 0.01 mg/g β-CN was released, equivalent to 0.02 mg/200 mL of milk. It could be concluded that between A1A1 and A2A2 β-CN, the release rate of BCM-7 is higher after the consumption of A1A1. In 2019, Asledottir et al. [11] conducted an ex vivo study using gastrointestinal juice from healthy humans to increase the physiologic relevance of the model compared to previous models using artificial or animal gastrointestinal fluids to analyse the presence of BCM-7 following in vitro digestion. However, this study did not evaluate the BCM-7 release rate.

The in vitro study of the Nguyen research group [48] also showed the rate of release of BCM-7 in the amount of 4,94 to 7,70 ng/mg/protein in A1A1 milk, but has not identified the release of BCM-7 following the digestion of A2A2 milk. The detection of BCM-7 was performed using ultra-high-performance liquid chromatography-tandem mass spectrometry (UHPLC-MS/MS) and subsequently confirmed using ultra-high-performance-high-resolution liquid chromatography orbitrap mass spectrometry (UHPL-HRMS). The difference in this study was the use of heat treatment, with temperatures and times similar to those used in pasteurizing, in milk samples prior to in vitro digestion.

Two other research groups also quantified the release of BCM-7 in vitro. Haq et al. [49], quantified the release of BCM-7 after the digestion of A1A1, A1A2 and A2A2 milk using the ELISA method. In this study, the values found for BCM-7 were 0.20 mg/g β-CN in A1A1 milk, the amount detected after A1A2 milk digestion was 0.06 mg/g β-CN, and in A2A2 milk, BCM-7 was not detected. The most recent study was performed by Cattaneo et al. [50], in which BCM-7 in A1A1 and A2A2 milk was quantified after in vitro digestion. The amount of peptide detected in A1A1 was 4 mg/g β-CN, similar to that found in the Haq et al. 2015 study, while the peptide release rate was 1.4 mg/g β-CN in A2A2 milk.

The methods used to quantify the release of BCM-7, both in in vitro and ex vivo studies, are not able to determine a minimum BCM-7 level with the potential to result in pharmacological effects in the gastrointestinal tract, and although they are easier, less expensive and faster, they do not correspond to the human physiology; in addition, parameters or specific individual differences are not considered. Therefore, data extracted from these studies are inconclusive and limited to fully support the possible adverse health effects of BCM-7 [16]. Thus, the in vivo method using animals to verify possible local and systemic effects of BCM-7 is still used as a human parameter model.

Few studies using animal models have established the dose/effect factor under BCM-7 intake. In the study with mice performed by Haq, Kapila, and Saliganti [6], 7.5 × 10^−8^ mol/day/animal of synthetic BCM-7 was employed in 200 μL of phosphate buffer saline solution. This amount was calculated based on the human dose translation. In this work, inflammatory effects were reported in the gut of mice associated to exposure to BCM-7 doses. In contrast, a recent study conducted by Yin et al. [51] verified the effect of different BCM-7 doses on the intestinal mucosa of older mice; three treatments were performed: Group I received a dose of 2 × 10^−7^ mol/day, which is considered a low BCM-7 dose; Group II received 1 × 10^−6^ mol/day (intermediate dose); Group III received 5 × 10^−6^ mol/day (high dose). In this study, it was reported that the low BCM-7 dose reduced proinflammatory factors, a positive result considering the age of animals, while intermediate and high BCM-7 doses indicated an increase in immune cells and local oxidative stress.

The in vivo methodology with animals is widely used to perform in situ investigations, which is a method that is closer to the human model compared to other options, but this is a method considered to be expensive, time consuming and with low correlation [46]. Despite these challenges, animal, in vitro and ex vivo studies can be used as a basis for future adaptations in human trials.

In an in vivo study in healthy humans, the amount of BCM-7 released after the ingestion of 30 g of A1 β-CN diluted in 500 mL of shakes for 7 days was determined, with individuals equipped with a nasogastric tube that migrated to the proximal jejunum after intervention. Then, it was possible to quantify the presence of 4 mg of accumulated BCM-7 in the jejunal effluent [39]. This amount is close to the proportion of BCM-7 of milk containing the A1A1 genotype after ex vivo digestion quantified by Asledottir et al. [3]. From this result, it was suggested that this amount is capable of causing biological action. However, this study did not evaluate possible local effects on the gastrointestinal tract due to several reported difficulties with nasogastric tube tolerance such as pain, vomiting and the absence of migration through pylorus [3]. In any case, this study was essential to confirm the evidence of the release rate of BCM-7 in in vitro and ex vivo studies. Similar studies should be carried out to more clearly understand the correlation of BCM-7 under local physiological effects; in addition, the use of milk in the intervention should be considered, since the digestion of other milk proteins can raise questions about interference in the kinetics of β-CN hydrolysis.

Another in vivo study in humans that also found potential results on the in situ effect of BCM-7 was conducted by the Sun et al. [7] research group, which evaluated through “Smart Pill” the gastrointestinal function of individuals with lactose intolerance after the consumption of 500 mL of A1A2 milk compared to the consumption of A2A2 milk for 14 days. A relationship was observed in the time of gastrointestinal transit and inflammation of the intestine after the consumption of milk containing the A1 β-CN variant compared to the consumption of milk with only A2 β-CN. The observation of these adverse effects was correlated with the release of BCM-7, but the peptide was not quantified. In this study, the effects were associated with lactose intolerance, so the condition of these individuals was already considered an intestinal inflammation comorbidity, which can contribute to adverse responses; thus, the BCM-7 dose released from the hydrolysis of β-CN containing the A1 phenotype and the time of exposure of individuals to the peptide raise the possibility of causing biological effects. This result is consistent with the findings of Sheng et al. [8], in which exposure to 3.15 ng/mL of BCM-7 accumulated 5 days after intervention with milk containing A1 β-CN was correlated with acute gastrointestinal effects on children with lactose intolerance, but the same treatment did not show significant effects on non-intolerant children.

Other authors also suggest the correlation between gastrointestinal discomfort due to milk intake in lactose-intolerant individuals and the consumption of BCM-7 given that these studies show significantly higher Bristol stool values after the consumption of A1 milk. Meanwhile, after the intake of A2 milk, improvements in stool consistency and abdominal pain in lactose-intolerant individuals were demonstrated [45,47].

In addition, the dose/effect variable of BCM-7, both local and systemic, should be considered to establish possible health effects, since milk-derived peptides must resist both intestinal and plasma peptidase, and the magnitude of the effects depends on the extent of their absorption to cause biological effects. Moreover, other variables should be considered; for example, the possibility of changes in the release rate of BCM-7 after heat treatment of milk, as well as the different forms of processing for dairy production, since some studies have shown changes in the release rate of the BCM-7 peptide [35,52,53].

The studies reviewed here suggest that the dose of 1 × 10^−6^ mol/day BCM-7 in the range and/or from the release of 0.06 mg/g A1 β-CN may be the minimum exposure threshold for triggering biological effects [48,49,51]. Other issues should be considered regarding the negative potential of BCM-7, since possible effects are not correlated with the entire population, which requires further investigation of its relationship with human health. This fact has deepened investigations, leading to the conclusion that not all individuals are affected by the peptide, since this can occur only in a small portion of the population. Usually, these individuals are considered more susceptible to exposure to and action of BCM-7, which can trigger symptoms of various forms, individual or concomitant to other health conditions [43]. Therefore, more studies, especially with humans, are needed to answer this and several other questions.

## 5. Evidence from Animal and Human Studies on the Potential Adverse Effects of BCM-7 on the Gastrointestinal, Immunological, and Neurological Tract

### Animal Studies

Some animal studies have been conducted to verify the possible effects of BCM-7 directly on the gut. The main findings relating the effects of BCM-7 to the gastrointestinal tract were delayed intestinal transit, increased inflammatory and immunological markers, increased microbiome stress, exaggerated increase of mucus. With the advancement of animal studies, interactions of BCM-7 with the redox system could also be demonstrated, which generates another field to be explored (Table 2).

There is scientific evidence that β-Casomorphins are μ-opioid receptor agonists [12]. This evidence can be observed in parts when there is sign of modulation in the response of the gastrointestinal transit time mediated by opioid receptors. The transit time in the intestine exclusively of β-CN is around 6 h in studies that evaluated the digestion time of milk containing A1 β-CN [5,39]. In the study by Daniel et al. [5] with rats that ingested naloxone, a specific opiate receptor antagonist before casein consumption, gastrointestinal transit time reduced by approximately 1 h was observed using feces as a measure. More compelling evidence was presented in the study of Barnett et al. [12], in which the comparison of consumption of milk containing A1 β-CN by rats significantly decreased gastrointestinal transit compared to consumption of milk containing only A2 β-CN (*p* = 0.03). In this study, the synergistic effect of naloxone was also evaluated, and the effect on the gastrointestinal transit time was significantly reduced after treatment with milk containing A1 β-CN (*p* = 0.01), but had no effect on time when milk with A2 β-CN was consumed exclusively. However, it is important to note that none of these studies evaluated the opioid effect of BCM-7 alone.

Another way to evaluate the opioid nature of the BCM-7 effect was first described by Claustre et al. [57] under the modulation of mucin release at the intestinal lumen. Mucins are the main protective components of the physiological gastrointestinal defense mechanism, secreted by goblet cells. In this study, mucin production in the jejunum of perfused rats was evaluated after luminal administration of isolated casein hydrolysate versus casein hydrolysate after naloxone administration and luminal BCM-7 and BCM-7 administration after naloxone administration. Both in the casein hydrolysate and BCM-7 administration, mucin discharge was significantly higher when not associated with naloxone (*p* = 0.05). On the other hand, the most recent work on the increase in mucin release under the effect of casein peptides was carried out by Fernández-Tomé et al. [55], in which the release rates of MUC 2 and MUC 3 mucins were higher in rats fed with casein hydrolysate compared to those of rats in the control group, but this study did not evaluate the isolated effect of BCM-7. However, excessive production of mucin may disrupt gastrointestinal function and harm commensal bacteria [40].

The relationship between β-Casomorphin digestion and the triggering of protective responses is correlated to the fact that the epithelium of the gastrointestinal tract, in addition to providing a physical barrier, also contains several different cell types, such as enterocytes, secretory cells, chemosensory cells and lymphoid tissue, which together are responsible for immune response, release of pro-inflammatory cytokines, mucus and neuroendocrine compounds [58]. If the possible effects on the release of mucins suggest a potential response in the protection of the gastrointestinal mucosa to the release of BCM-7, this statement still lacks further studies and clarifications, since there is a shortage of studies considering the direct effect of BCM-7.

As previously mentioned, BCM-7 has the potential to activate immune-inflammatory cells. In the work of Haq et al. [6], the ingestion of BCM-7 by mice in a 15-day period, used as an experimental model of human applicability, demonstrated an increase in the amount of inflammatory molecules, myeloperoxidase (MPO), IL-4 and histamine in intestinal fluid samples, the molecules of which have the ability to cause tissue damage, such as acute or chronic inflammation, and also increased humoral immune response through the following biomarkers: IgE, which can be manifested by various allergic reactions, including food allergy; and IgG, suggestive against food antigens for causing low-grade inflammation in the intestinal mucosa. The increase in MPO as a neutrophil activation marker, indicative of inflammatory response, is consistent with the findings also reported in the study by Barnett et al. [12].

In the study by Yin et al. [51], the BCM-7 effect/dose in the intestinal mucosa of rats increased the immunoglobulin A (IgA) levels, an important indicator of immune function; in addition, an increase in both antioxidant and oxidant cytokines was observed, which led to the imbalance of the redox state, elevating the local oxidative stress condition. Certainly, increased oxidative stress promotes the release of pro-inflammatory cytokines, which enhance the effects of gastrointestinal inflammation [37]. In contrast, a previous study of the same group [59] showed that exposure to BCM-7 in rats with diabetes resulted in a protective effect on the pancreas, reducing oxidative stress through increased levels of enzymatic antioxidant superoxide dismutase (SOD), catalase (CAT) and glutathione peroxidase (GSH-Px). In the work of Han et al. [60], positive association with the release of BCM-7, beneficial effects on increased antioxidants SOD, CAT and GSP-Px and reduction in the oxidative stress in myocardium in diabetic rats were observed, also associated with reduction in hyperglycaemia. However, in a more recent study, the effect of consumption of A1 or A2 casein hydrolysate by chemically induced diabetic adult rats had no significant effect on pancreas, kidney, liver or cardiac function [61]. Despite this, previous studies have associated the consumption of A1 β-CN with diabetogenic conditions in rats compared to A2 β-CN, but this is not clearly proven [56,62]. This suggests that differences in health conditions may trigger different physiological responses under the effect of BCM-7.

The use of experimental in vivo animal models does not always simulate real human conditions, especially when associated with diseases, since there are physiological, biochemical and anatomical differences between species, which limits conclusions [46].

Associations have also been made between β-Casomorphins and behavioural effects of analgesia, which is the reason that led the research group of Sun and Cade [63] to conduct studies with rats in the behavioural field, the results of which uncovered relationships with autism and schizophrenia in humans. Even in clear conditions that BCM-7 activated brain regions similar to those affected by schizophrenia and autism conditions, the certainty that these areas would also be stimulated under these conditions in humans has not yet been studied. This is an unexplored field, and scarce information is available in the literature.

Of all the possible effects of BCM-7 already studied in animals, the relationship with the gastrointestinal system and the possible modulations of local responses are those that present the most compelling information. In addition, some questions were not possible to be answered in animal studies. As already reported, for BCM-7 to trigger local effects at the gut level, it is necessary to resist the first biological barrier, the hydrolysis of DPP-IV. However, studies with animals have failed to establish the relationship between the action of DPP-IV and the cleavage of BCM-7. According to Barnett et al. [12], there is positive feedback from DPP-IV in response to BCM-7 release after the consumption of A1 β-CN. However, further studies are needed to compare the effects of A1 and A2 β-CN on binding to DPP-IV, as well as the hydrolysis of BCM-7 to other peptidases.

The recent study by Guantario et al. [54] included milk in the diet of older mice and found results different from those obtained by Barnett et al. [12]. In this study, there was no significant difference in DPP-IV, MPO and several other proinflammatory cytokines between A1A2 and A2A2. In contrast to previous studies, no change was observed in the level of intestinal inflammation by dairy supplementation. However, the A2A2 group showed better morphology of intestinal villi, which is suggestive of the fact that it can partially neutralize the effect of aging.

Although animal trials help science to understand biological issues, the barrier between animal and human physiology does not always allow clarifying some points; for example, the reason that only part of the population is more likely to experience adverse effects from BCM-7, while another portion of the population experiences potential health benefits. These responses are difficult to obtain because they require in-depth studies that consider the intestinal microflora and association with individual health conditions [39,64].

## 6. Human Studies

Some studies have been conducted in humans in order to investigate the possible effects of BCM-7. One of the reasons why there are few reports in the literature is the fact that in situ investigations are complex and difficult to perform. The use of invasive exploration equipment can cause discomfort to volunteers, and the equipment also increases the cost of studies. Further, concerns about the potential risks of clinical trials also limit the volunteer engagement. Therefore, most studies associate the symptomatologic responses of the gastrointestinal tract to the levels of serum and urine biomarkers (Table 3).

Three of the human studies tested and quantified BCM-7 and DPP-IV in systemic markers in autistic children. It is important to highlight that in these studies, no dietary intervention was performed in the different groups, and in some of them, only nutritional information was collected from children (Table 3). The latest research found significantly higher amounts of BCM-7 and DPP-IV in serum in the autistic group than in neurotypical children (*p* < 0.001) using the commercial ELISA^®^ kit [43]. Higher BCM-7 levels in autistic children compared to neurotypicals were also reported in the study by Sokolov et al. [65], who verified the presence of the peptide in urine using ELISA as quantitative method. However, this study did not investigate the relation of BCM-7 with DPP-IV. The correlation between BCM-7 and the DPP-IV enzyme was studied by a Polish research group in 2015 [66]. In this study, two groups of children were examined, autistic and neurotypical, with promising results, although the relationship between BCM-7 and DPP-IV was explored through peripheral blood mononuclear cells in an in vitro study. It was possible to identify similar DPP-IV gene levels under the influence of BCM-7 and hydrolyzed milk in both groups, but there was no expression of the µ-opioid gene under the influence of BCM-7.

Jarmołowska et al. [43] suggested that it is likely that the considerable increase in BCM-7 increased the DPP-IV levels in the blood to cleave the peptide, although this mechanism was not effective in keeping opioid peptide levels high in the bloodstream. However, negative feedback in this sense has already been reported. Pathological factors can also initiate the suppression of brush border enzymes, particularly DPP-IV, as reported by Cieślińska et al. [66]. DPP-IV deficiency can occur mainly in individuals with chronic disease or psychiatric disorders, as they are health conditions associated with compromised immune status, as well as in individuals predisposed to gastric disorders, such as poor digestion and abdominal discomforts [67]. The understanding of health effects of food-derived exorphins is complicated due to individual characteristics such as genetic susceptibility, specific health conditions and different interactions between them, which leads to some confused results in different trials.

Another method combines dietary intervention with gastrointestinal symptoms. Two recent studies concluded that in lactose-intolerant patients, after consuming A2A2 milk, gastrointestinal discomforts were reduced compared to the case of conventional milk (A1A2) consumption; and symptoms of pain and fecal urgency were the main findings [45,68]. In this sense, He et al. [69] observed acute gastrointestinal symptoms associated with milk consumption, such as abdominal pain, bloating, stool frequency and consistency measured by Bristol Stool Scale after the consumption of conventional and A2A2 milk after 1 to 12 h by lactose-intolerant individuals. Among the results, it was observed that abdominal pain, distension and stool frequency were reduced after the consumption of A2 milk for 12 h by lactose-intolerant individuals. Those studies found a correlation between improved gastrointestinal discomfort and reduced exposure to BCM-7 through the consumption of A2A2 milk, although the peptide was not quantified in these studies. These suggest that A2A2 milk may result in higher tolerance to dairy consumption.

**Table 3 foods-12-03151-t003:** Recent in vivo studies with humans on the potential effects of BCM-7 on discomfort and immune–inflammatory responses in the gastrointestinal tract.

MODEL	Design	Methods	Biomarkers Evaluated	Sample	Significant Results/Outcomes	Study
**Children** **3–10 y** **(n = 137)**	51 Neurotypical group 86 ASD group	No dietary intervention	DPP-IV and BCM-7	Blood and urine	Levelng/mL	ASD	Control	Jarmołowska et al., 2019 [43]
BCM-7 blood	42.9 ± 2.5	26.4 ± 1.6
DPP-IV blood	1089 ± 44	934 ± 52
No significant difference was detected in BCM7 urinary levels
**Children** **4–8 y** **(n = 20)**	10 Neurotypical group 10 ASD group	No dietary intervention	BCM-7	Urine	Levelpg/mL	ASD	Control	Sokolov et al., 2014 [65]
BCM-7 urine	75 ± 10	58 ± 7
**Children and young** **3–19 y** **(n = 404)**	296 Neurotypical group 88 ASD group	No dietary intervention	μ-opioid receptor and DPP-IV gene expression	Blood	Correlation in DPP-IV gene expression under the influence of BCM7 and hydrolyzed milk among healthy children and ASD with genotype GG.	Cieślińska et al., 2015 [66]
**Chinese lactose-intolerant children** **5–6 y** **(n = 75)**	A1A2 milk group A2A2 milk group	150 mL twice daily, 5 days of intervention; Double-blind, randomized and crossover	Gastrointestinal discomforts, Cognitive responses, BCM-7, MPO,IgE, IgG, IgG-1, IL-4 and GSH	Blood andfeces	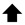 IgE, IgG, IgG-1 and IL-4 in the A1A2 milk group 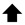 GSH in the A2A2 milk group 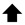 BCM-7 in the A1A2 milk group 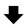 frequency of stool in A2A2 milk groupImproved consistency of stool in the A2A2 milk groupImprovements in accuracy of the Cognitive Impairment Test in the A2A2 milk group	Sheng et al., 2019 [8]
**Lactose-intolerant adults of various ethnicities** **(n = 25)**	A2A2 groupA1A2 group (25%A1/75%A2) A1A2 group (75%A1/25%A2)Lactose-free A1A2 group	245 mL single-meal; Double-blind, randomized and crossover	Gastrointestinal symptoms over 6 h		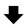 Abdominal pain in the A2A2 group	Ramakrishnan et al., 2020 [45]
**Chinese lactose-intolerant** **adults** **(n = 600)**	A1A2 milk group A2A2 milk group	300 mL single-meal;Double-blind, randomized and crossover	Gastrointestinal symptoms over 1 h, 3 h and 12 h	Feces	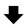 Abdominal pain, borborborygmus, flatulence, bloating, frequency and consistency of stool after 3 h in the A2A2 group	He et al., 2017 [69]
**Australian adults** **(n = 37)**	A1A1 milk groupA2A2 milk group	750 mL/dia, 2 weeks of intervention; Double-blind, randomized and crossover	Symptoms and transit time gastrointestinal	Feces	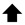 Results of stool consistency (Bristol Stool Scale) in the A1A1 milk group	Ho et al., 2014 [47]
**Chinese lactose-intolerant adults** **(n = 45)**	A1A2 milk groupA2A2 milk group	250 mL twice daily 2 weeks of intervention;Double-blind, randomized and crossover	Symptoms and transit time gastrointestinal, GSH, MPO, IgG, IgG, IgE, IL-4 and PCR	Gastrointestinal photograph, blood andfeces	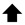 IgE, IgG, IgG-1 and IL-4 in A1A2 milk group 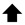 gastrointestinal transit time A1A2 milk groupA2A2 milk intake was not associated with increased gastrointestinal symptomsPCR did not differ between groups	Sunet al., 2015 [7]
**New Zealander lactose-intolerant women** **(n= 40)**	A1A2 milk groupA2A2 milk groupLactose-free A1A2 group	750 mL single-meal;Double-blindand randomized	Gastrointestinal symptoms over 3 hBowel movements		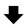 Symptoms of nausea and fecal urgency in A2A2 and lactose-free milk groupBowel movement frequency does not differ between groups	Milan et al., 2020 [68]
**Children with functional chronic constipation** **1–12 y** **(n = 39)**	A1A2 milk group A2A2 milk group	400 mL/dia, 2 weeks of intervention	Gastrointestinal discomforts	Blood andfeces	There was no effect of the casein type	Crowley et al., 2013 [70]

For all in vivo studies with animals the value of *p* < 0.05 was considered. DPP-IV = Dipeptidyl peptidase IV; BCM-7 = β-Casomorphin-7; ASD = Autism Spectrum Disorder; PCR = C-Reactive Protein; y = Years old; MPO = Myeloperoxidase; GSH = Glutathione.

In addition to the inflammatory effect by exposure to BCM-7 reported in these studies, the effect on bowel propulsion seems to be a predominant characteristic [7,12]. The reduction in motility consequently causes delays in gastrointestinal transit [12]. Interestingly, µ-opioid receptors are highly expressed in the small intestine, with the ability to regulate and coordinate peristalsis, as well as secretion from the gastrointestinal tract [10]. BCM-7, when binding to μ-opioid receptors in the gastrointestinal tract, in addition to modulating the mechanics of intestinal propulsion resulting in reduced motility, also deregulates mucus and hormone production [40]. Additionally, increased mucus secretion leads to gastrointestinal damage when this increase is exaggerated [40].

Remarkably, Ho et al. [47] also identified an increase in local discomforts in healthy individuals after the consumption of A1A2 milk (without lactose intolerance) when compared to a group that consumed A2A2 milk. Significant changes in stool consistency were observed between groups, as well as a significant positive correlation between abdominal pain and softer stool after the consumption of A1A2 milk (*p* < 0.001), raising the hypothesis that these adverse factors were induced by proinflammatory cytokines [47]. However, this study has limitations, and further studies using the same model with a higher number of individuals and with more clinical analyses are necessary to more clearly understand these effects.

In contrast, the study conducted by Crowley et al. [70] showed no significant difference in patients after the intake of 400 mL of A1A2 and the same amount of A2A2 milk. A two week period was considered in children with chronic constipation, the measurements of which were performed by bowel movements in relation to transit time. Therefore, under these experimental conditions, the different milk genotypes were not significantly associated with better resolution of chronic constipation symptomatology [70].

Dietary clinical trials that provide an assessment of intestinal and systemic biomarkers may lead to more accurate responses on the action of BCM-7 in humans. Based on these parameters, only two studies were identified in the literature reviewed, namely the studies of Sun et al. 2015 and Sheng et al. [7,8].

The study by Sun et al. [7] greatly contributed to associate the in situ inflammatory process by exploring the entire gastrointestinal tract with the “Smart Pill” in individuals who consumed A1A2 milk compared to lactose-intolerant (IL) or tolerant (control) individuals that consumed A2A2 milk, characterized as a double-blind, randomized, crossover study with 45 Chinese individuals. Thus, it was possible to confirm that under experimental conditions, the gastrointestinal transit time was reduced after the consumption of A1A2 milk by lactose-intolerant individuals. The reduction in the gastrointestinal transit time is supported by the increase in inflammation by 36.4% in the small intestine, as well as an increase in stomach inflammation by 22.7% in the same individuals compared to the control group, considering as significant values *p* < 0.05. These data were also reinforced by the systemic increase in serum biomarkers such as IL-4 and Immunoglobulins (IgG, IgE and IgG1) in the IL group.

The most recent study was conducted by Sheng et al. [8], who supervised a double-blind, randomized and cross-reference dietary intervention in children from age five to six. In the clinical trial, Chinese infants suffering from lactose intolerance consumed 300 mL/day of A1A2 or A2A2 milk, for five days. An increase in the same immunologic and inflammatory biomarkers (IL-4, IgG, IgE and IgG1) was shown, demonstrated by Sun et al. [7], as well as changes in the MPO marker associated with the consumption of conventional milk (A1A2) by lactose-intolerant patients compared to similarly intolerant patients who consumed A2A2 milk; the latter group experienced an increase in GSH levels. Quantification of BMC-7 was important for this clinical trial as well. In the results, it was possible to observe an increase in peptide in the serum of children who consumed A1A2 milk in comparison to the group who consumed A2A2 milk. Changes in gastrointestinal parameters were observed in children who consumed A2A2 milk, including a reduction in frequency of and an improvement in stool consistency [8].

In particular, the results of the few human studies carried out so far emphasize the hypothesis that the adverse effects of BCM-7 can be attributed in principle to a small portion of the population, considering susceptibility factors such as intestinal microbiota, permeable intestinal barrier, altered or immunocompromised immune response, and DPP-IV deficiency [71]. Some comorbidities resulting from non-communicable diseases or neurological disorders can lead to this susceptibility.

As in animal studies, human studies highlight findings relating it to the potential effects of BCM-7 on the gastrointestinal tract such as gastric disorders and abdominal discomfort similar to the effects caused by lactose intolerance, increased mucus secretion, reduced motility, pain and abdominal distention (Table 3).

In addition to the relationship with the gastrointestinal system, studies also considered the consumption of milk of different genotypes and correlation with Type 1 Diabetes, as performed by the research group of Elliott et al. and Bell et al. [72,73]. However, this was not conclusive, and the topic requires further investigation, since these studies used ecological design. Chin-Dusting et al. [64] investigated, in a double-blind crossover study, the relationship between A1 β-CN and increased cardiovascular risks in individuals at a high risk of developing cardiovascular diseases, and comparisons between daily supplementation with A1 or A2 casein for these individuals did not result in statistical differences on measures related to the pathology.

Overall, the few human studies do not provide conclusive results that confirm the adverse effects of BCM-7 due to milk consumption. The studies reported here showed the potential of BCM-7 to promote greater gastrointestinal discomfort and increased levels of some immune–inflammatory markers as observed in animal studies. In addition, a gastrointestinal inflammation condition is suggested, which can trigger secondary effects in the abdominal region such as pain, distension, flatulence, and diarrhea. Thus, the consequences of symptoms are reflected in the CNS and lead to the belief that they can alter behaviour responses, information processing and neural connectivity [74].

## 7. Gut–Brain Axis

Recent studies have increasingly focused on the influence of the intestinal tract and its connection on the brain. The gut–brain axis refers to the mechanism of bidirectional communication between the gastrointestinal tract and the central nervous system (CNS) [58]. Changes in this axis can be caused by local effects such as malabsorption of nutrients required for good brain function, or by secondary activation, causing gastric discomforts and pain that consequently result in a stress condition. In addition, they promote the activation of the immune and endocrine system, and increase oxidative stress. Communication can also be performed by neural pathways even by small molecule messaging systems [7]. It is proposed that these effects have the ability to interfere with the development of the human brain, behaviour, cognition and mood [41,75].

Due to the large innervation of the gastrointestinal tract, ingested components can send signals to the CNS, and communication can be established directly through the bloodstream [74]. This direct communication of the gut–brain axis allows the intervention of the molecules in the modulation of responses in the CNS, and therefore, exorphins derived from food, especially dairy and gluten, and their potential effects on health problems have been continuously investigated, which has enhanced arguments in favour of gluten- and casein-free diets (GFCF) [32].

Although BCM-7 is considered a large peptide containing seven amino acids, its absorption and transport through the intestinal epithelium occurs when there is DPP-IV deficiency, the evidence of which has already been reported in in vitro and in vivo studies with animals and humans [6,43]. In addition, the hypothesis that the condition of increased intestinal permeability is associated with chronic diseases or neuropsychiatric disorders has also been defended, which may facilitate the passage of the peptide into the bloodstream [37]. In individuals with specific health conditions, DPP-IV deficiency together with increased intestinal permeability are crucial factors that might facilitate the passage of BCM-7 through the epithelial barrier.

The absorption and transport of long peptides such as BCM-7 in infants aged 6–12 months has already been reported, and there are a variety of studies that have analysed its effects on these individuals, a fact that occurs because the gastrointestinal tract is immature and consequently more permeable, which facilitates the passage of long peptides that escape from hydrolysis even when there is no DPP-IV deficiency [53].

If the main dairy Casomorphin, BCM-7, is able to reach the bloodstream and different tissues and organs, it has not yet been proven in humans, requiring further studies. What is much discussed is its potential to bind to the μ-opioid receptor. The main question is whether the widespread presence of opioid receptors beyond the intestine, in the brain and various other organs. BCM-7 appears as a direct consequence to trigger systemic effects, mainly neurological effects of analgesia and behavioural change. However, multiple influencing factors can lead to a great diversity of effects among individuals [32,58].

Although this hypothesis is not conclusive for humans, in animals, there are some indications. A recent study in rats conducted by Osman et al. [76] explored mood changes in rats after the consumption of A1A2 and A2A2 milk. The results concluded that there was a significant increase in stress-induced immobility in mice fed with A1A2 milk, which is reinforced by specific changes in the brain region of μ-opioid receptors, oxytocin and changes in urinary profiles. In addition, another important finding was the increase in intestinal microbiota after the intervention with A1A2 milk but not with A2A2 milk [76]. There are μ-opioid receptors throughout the CNS located in the brainstem, bulb, spinal cord, hypothalamus, and the limbic system, but they are also found in the peripheral nervous system [10].

Studies on the action of BCM-7 in the human CNS reported the activation of the gut–brain axis due to gastric discomfort, and not directly by the binding of BCM-7 to µ-opioid receptors in the CNS. This relationship is evidenced in the study conducted by Sheng et al. [8] where, after the consumption of A2A2 milk by school-aged children with lactose intolerance, the subtle cognitive impairment test showed improvements in response accuracy and greater processing efficiency. This association with the cognitive system was correlated with the reduction in the effects on the gastrointestinal system caused by BCM-7 released due to the consumption of A1A2 milk. This is a potentially important result, in which exposure to BCM-7 briefly causes side effects in the CNS, since direct consequences of BCM-7 to the CNS or other organs are difficult to investigate.

Another potential emerging field is the possibility of BCM-7 indirectly influencing the CNS through stress to the intestinal microbiota. The gut–brain axis theory was expanded to the microbiota–gut–brain axis to explain the influence of microorganisms on cognitive development and functioning [37]. The intestinal microbiota is the community of symbiotic and pathogenic microorganisms present in the intestine, and this microbiota is critically involved in the communication of the intestine–brain axis, influencing the brain and behaviour modulation, also presenting analgesia effects (Figure 3) [58].

Importantly, microbiota disorders may also be associated with excessive mucin production in the intestine triggered by BCM-7 [77]. Intestinal microbiota disorders can influence the CNS by the deficiency in the absorption of molecules used as a substrate for its proper functioning, for example, the absorption of tryptophan, precursor of serotonin, an important neurotransmitter in the CNS involved in the regulation of mood, appetite, memory, learning and sleep [75]. These data reinforce the hypothesis raised by Osman et al. [76], who observed changes in the microbiota of mice accompanied by changes in mood through gut–brain communication, suggesting that this effect was due to the effect of BCM-7 released from the consumption of A1A2 milk.

Some diseases and disorders present as characteristics the imbalance of intestinal microorganisms, such as in gastrointestinal disorders, celiac disease, obesity, diabetes, as well as in mental disorders, eating disorders, autism spectrum disorders and mood disorders [75]. Certainly, consideration of the interference in the microbiota is crucial to understanding changes in the gut–brain axis.

## 8. Concluding Remarks and Future Directions

The findings of bioactive peptides with potential health benefits have been the subject of increasing interest in the context of functional health-promoting foods, as well as the possible adverse effects from milk consumption based on hypothesis A1 and the release of BCM-7 [6].

In general, adverse reactions to milk consumption are usually referred to as lactose disaccharide intolerance [77]; however, recent studies have elucidated the hypothesis that gastrointestinal discomforts can be exclusively attributed to the release of BCM-7 from the hydrolysis of A1 β-CN, especially when associated with non-communicable diseases and neurological disorders [7,43,47]. Some populations in different age groups with these conditions have been studied, such as children of preschool age and adults, and, with less frequency, healthy individuals with or without additional complaints of gastrointestinal nature, all these studies seeking to understand and identify responses linked to the action of BCM-7 [77].

Among the groups of individuals considered more susceptible, the following parameters stand out: Lactose intolerance, autism, type 1 diabetes, cardiovascular diseases, sudden infant death syndrome and chronic childhood constipation [10,16,77]. Diet can play a key role in the exacerbation of symptoms related to these health conditions [77], although the European Food Safety Authority (EFSA), after a review of papers published until 2009, published an opinion informing the population that no clear evidence of a cause/effect relationship was found between BCM-7 and the development of some non-communicable diseases. However, EFSA subsequently concluded that the reviewed studies were not sufficient to claim the A1 variant as a cause or risk factor for the onset of these diseases, but reported that β-Casomorphins may cause gastrointestinal tract disorders, such as bloating, flatulence and abdominal pain. In addition, there is need for further studies to verify the implication of the A1 variant in the development or worsening of the symptoms of non-communicable diseases [10].

Even after the Opinion of EFSA in 2009, recent studies emerged with groups of individuals considered by this review as more susceptible to the potential effects of BCM-7 (lactose intolerance, autism, type 1 diabetes, cardiovascular diseases, sudden infant death syndrome and chronic childhood constipation). However, these studies are scarce, while some approaches remain unexplored.

Therefore, due to the few studies in humans, results from clinical trials are inconclusive [10]. Evaluation of different health conditions makes these studies even more difficult to be carried out, in addition to the various difficulties for conducting in vivo studies [32]. Although clinical trials have not yet been sufficient to establish a clear relationship of the adverse effects of BCM-7 on different physiological responses, the likelihood that this mechanism may initiate or exacerbate some gastrointestinal symptoms seems to be high, the evidence of which has been reported in most studies and explored in this review.

In summary, after elucidating the main challenges of establishing the adverse effects of BCM-7 such as delimiting minimum BCM-7 content, difficulties of clinical trials in humans, mainly in groups associated with some non-communicable diseases and neurological disorders, other factors interfere with more conclusive results, for example, inadequate clinical analyses, short intervention trials and insufficient number of participants. Research is expected to continue to evolve as new studies are being developed and more results are released in the very near future.

## Figures and Tables

**Figure 1 foods-12-03151-f001:**
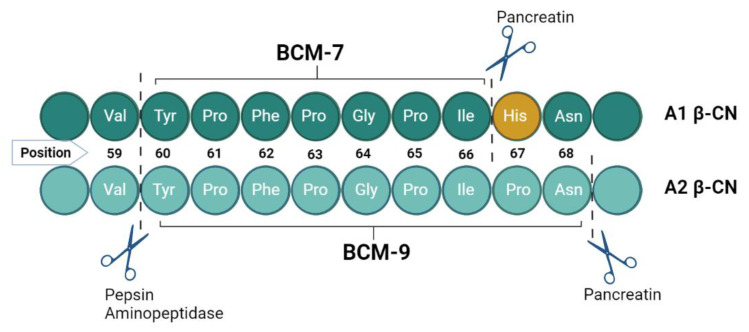
Peptide binding of A1 and A2 β-CN and cleavage sites with BCM-7 and BCM-9 release during proteolysis.

**Figure 2 foods-12-03151-f002:**
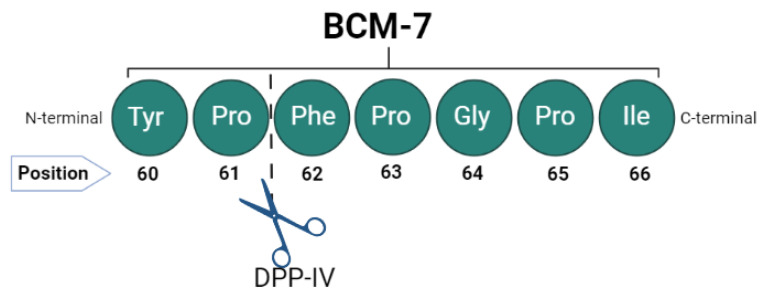
Cleavage of BCM-7 between proline and phenylalanine residues in the peptide.

**Figure 3 foods-12-03151-f003:**
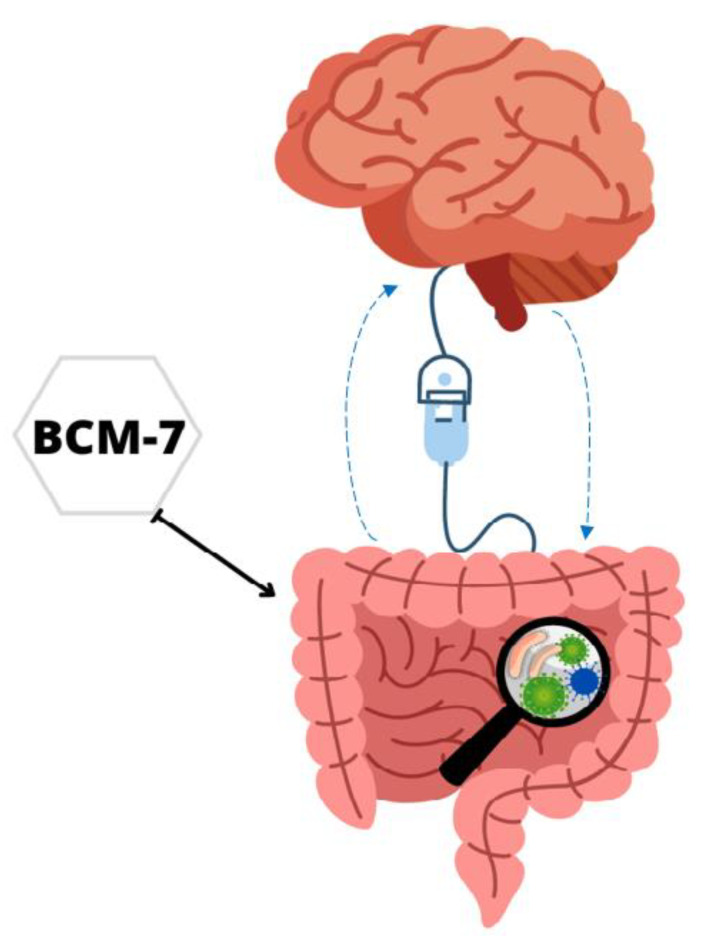
Potential interaction of BCM-7 in the microbiota–gut–brain axis.

**Table 1 foods-12-03151-t001:** Sequence of peptides that form β-Casomorphin (Adapted from Nguyen et al. [35]).

Source: β-CN Bovine	Peptide Sequence	Position
**β-Casomorphin-4**	**Tyr-Pro-Phe**-Pro	60–63
**β-Casomorphin-5**	**Tyr-Pro-Phe**-Pro-Gly	60–64
**β-Casomorphin-6**	**Tyr-Pro-Phe**-Pro-Gly-Pro	60–65
**β-Casomorphin-7**	**Tyr-Pro-Phe**-Pro-Gly-Pro-Ile	60–66
**β-Casomorphin-8**	**Tyr-Pro-Phe**-Pro-Gly-Pro-Ile-Pro	60–67
**β-Casomorphin-9**	**Tyr-Pro-Phe**-Pro-Gly-Pro-Ile-Pro-Asn	60–68
**β-Casomorphin-10**	**Tyr-Pro-Phe**-Pro-Gly-Pro-Ile-Pro-Asn-Ser	60–69
**β-Casomorphin-11**	**Tyr-Pro-Phe**-Pro-Gly-Pro-Ile-Pro-Asn-Ser-Leu	60–70

**Table 2 foods-12-03151-t002:** Recent in vivo studies with animals on the potential effects of BCM-7 on discomfort and immune–inflammatory responses in the gastrointestinal tract.

MODEL	Design/Intervention	Dosage/Length/ Administration	MeasuredBiomarkers	Samples	Significant Results/Outcomes	Study
**Male mice**	BCM-7 group BCM-5 group Control group	Exposure of BCM-7 or BCM-5: 7.5 × 10^−8^ mol/day/animal for 15 days intubated orally Exposure with saline phosphate buffer: 200 μL for 15 days intubated orally	MPO, MCP-1, IL-4, Histamine, IgG, IgG1, IgG2a, IgE, IgA andgoblet cells counting	Fluid and intestinal mucosa	Increase	BCM-7	BCM-5	Haq et al., 2014 [6]
MPO	129.76%	117.55%
MCP-1	33.38%	31.73%
IL-4	175.54%	164%
Histamine	167.59%	189.21%
IgE	77.09%	52.37%
IgG	42.13%	45.17%
IgG1	126.63%	159.78%
IgG2a	77.39%	90.27%
Compared to the Control groupThere was no difference in IgA and goblet cells.
**Male mice** **4 w** **(n = 48)**	A1S group = plus saline + milk powder with A1 β-CN A2S group = plus saline + milk powder with A2 β-CN A1N group = Naloxone + powder milk with A1 β-CN A2N group = Naloxone + powder milk with or A2 β-CN	A1 skim milk powder, 475 g total to 36 h or 84 hA2 skim milk powder, 468 g total to 36 h or 84 h	Gastrointestinal transit time, MPOAnd DPP-IV	Urine, feces,blood andintestinal mucosa	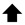 MPO activity between 64 and 65% in all groups, except in the A2N group 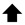 DPP-IV activity in 37 and 40% in A1N and A1S groups, respectively, compared to A2S group 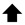 GI transit time was higher for the A1S and A1N groups	Barnett et al., 2014 [12]
**Elderly mice** **(n = 50)**	Control group young miceControl group age BCM-7 group (low, medium and high dose)	Control group young and aged mice: Exposure to saline physiological solution for 30 days through enteral administrationBCM-7 for 30 days through enteral administrationLow: 2 × 10^−7^ mol/day Medium: 1 × 10^−6^ mol/dayHigh: 5 × 10^−6^ mol/day	IL-2, TNF-α,IgA, SOD, CAT and MDA	Intestinal mucosa	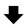 IL-2 level in the aged control group compared to the young control and low-BCM-7 groups 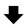 IgA level in the aged control group compared to the young control, medium- and high-BCM-7 groups 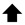 TNF-α level in the aged control group compared to the young control and low-BCM-7 groups 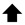 SOD in low- and medium-BCM-7 groups 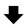 MDA: All BCM-7 groups	Yin et al., 2019 [51]
**Elderly mice** **(n = 24)**	Control groupA1A2 milk group A2A2 milk group	4 weeks of intervention120 g of lyophilized milk	MPO, DPP-IV, TNF-α, IL-6 and IgG	Feces, blood,fluid and intestinal mucosa	MPO, DPP-IV, TNF-α, IL-6 and IgG do not differ among the three groupsThe fecal microbiota found was different between A2A2 and A1A2 groups	Guantario et al., 2020 [54]
**Male rats** **8 w** **(n = 18)**	Control groupHydrolyzed casein group	8 weeks of interventionHydrolyzed casein group: 0.4 g hydrolysate/kg/day/ animal	Mucin, IL-6, TNF-α, glucagon, leptin and insulin	Feces andintestin	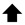 mucin excretion after 2 w (29%) and 8 w (47%) in Hydrolyzed casein groupIL-6, TNF-α, glucagon, leptin, and insulin no difference between the groups	Fernández-Tomé et al., 2017 [55]
**Young mice** **3–4 w**	A1 milk for five generationsA2 milk for five generations	The mice were fed ad libitum over a period of 30 weeks	Insulin, glucose, incidence of diabetes andimmune profile	Feces,blood,lymphatic tissue andIntestinal integrity	In the F3 generation, at 30 weeks, the incidence of diabetes was doubled after the intake of beta-casein A1 relative to the intake of A2 (A1: 40% vs. A2: 20.7%)In F4 mice, subclinical insulitis and glucose metabolism bias were evident in 10-week-old mice that only received A1 milk	Chia et al., 2018 [56]

For all in vivo studies with animals, the value of *p* < 0.05 was considered. MPO = Myeloperoxidase; MCP-1 = Monocyte chemotactic protein; DPP-IV = Dipeptidyl peptidase IV; IgA = Immunoglobulin A; SOD = Superoxide dismutase; MDA = malondialdehyde; w = weeks old.

## Data Availability

The data used to support the findings of this study can be made available by the corresponding author upon request.

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
