# Peer review of "Difficulties in Establishing the Adverse Effects of β-Casomorphin-7 Released from β-Casein Variants—A Review"

_foods, 2023, doi:10.3390/foods12173151_

Round 1

Reviewer 1 Report

As a scholar in a non-related field, I believe that this review is well written, comprehensive, and rigorous, which has taught me a lot of knowledge.
However, I think the author's description of the relationship betweenβ-CN variants (A1 variant?) and BCM-7 is slightly unclear.

Author Response

The authors of this review appreciate your comments and we are happy with them. About the relationship of A1 β-CN and BCM-7 not being so clear, we believe that this is one of the points we want attention in this review, since the topic can still be extensively explored. And so, we want to encourage new studies in the area, especially in vivo and in various populations.

Reviewer 2 Report

The authors have provided a comprehensive review on BCM-7 adverse reactions and the challenges for confirming such biological effects. The manuscript is interesting and well written. However, there are some remarks that should be taken into consideration.

1)  Typos: in table 2-footer, p symbol (of p value) should be italic as well as in vitro expression (please revise it in pages 6 and 7)

2) The authors should follow the journals style in reference citation (in the text and in the list of references)

3) Tables 2 and 3 should be improved since they are disorganized and misleading, making them difficult to follow.

4) Since BCM-7 has a negative impact on health, the authors should put an emphasis on the different methods used for the detection of BCM-7 in diary products.

Reviewer 3 Report

The review submitted by Marta et al seems good and written and presented in good way. I have totally in favor to publish in this journal after minor revision in data presented in tabular form. The tables should revised and presented data in good fashion. 

English is OK

Author Response

The authors of this review appreciate your comments and we are happy with them. We have made revisions and better organized the data presented in tables 2 and 3.

Reviewer 4 Report

The reviewed literature provides plausible indications of the hypothesis of a relationship between β-CN A1/BCM-7 and adverse health effects in vitro and in vivo clinical trials both in animals and humans. There are some format problems as following:

1. References in manuscript body. Page 2, Asledottir (not T. Asledottir), Waugh (not WAUGH), by Kruif and Holt (2003), β-CN by Aschaffenburg (1968); Page 3, Nguyen (not H. T. H. Nguyen); In whole manuscript, Sun et al. 2015 (not Jianqin et al. 2015); Page 6, Asledottir (not Tora Asledottir), Nguyen (not NGUYEN); Page 7, Haq (not Ul Haq), Yin (not YIN); Page 9, Yin (not YIN); Page 15, by Cieślińska et al. (2015), EI-alameey (not R EI-alameey); Page 17, Bell et al. (2006);.

2. About symbol. Page 1, β (not beta); Page 5, 30 min to 6 h; In Table 2, multiple sign is “×”, not x; decimal point is “.”, not “,”; 36 h, 84 h; in Male mice line, Haq et al. 2014 is coincide with data; In Table 3, mg/mL? or pg/mL? in lines 1 and 2, references are coincide with data; Significant results, Study; 1 h, 3 h,12 h;  

3. Others, Page 11, firstly (not first); Significant value is characterized by P or p? Uniform showing.

Minor editing of English language required

Reviewer 5 Report

The manuscript should be revised considerably.

The quality of English should be revised.

Round 2

Reviewer 2 Report

The authors indeed  have improved their manuscript and addressed most of my comments favorably. However, the list of references still need formatting into  journal's style. In addition some details are recommended regarding the detection of BCM-7 ( the authors could mention the role of LC-MS/MS, LC-HRMS analyses and Aptamers in the detection and sensing BCM-7).  I think theses papers may be useful:

https://doi.org/10.1016/j.jfca.2015.08.009

https://doi.org/10.1021/acs.jafc.5b00007.

Reviewer 5 Report

The manuscript has been revised significantly. All the comments have been incorporated.

No comments